# Assessment of the Methods Used to Develop Vitamin D and Calcium Recommendations—A Systematic Review of Bone Health Guidelines

**DOI:** 10.3390/nu13072423

**Published:** 2021-07-15

**Authors:** Zhaoli Dai, Joanne E. McKenzie, Sally McDonald, Liora Baram, Matthew J. Page, Margaret Allman-Farinelli, David Raubenheimer, Lisa A. Bero

**Affiliations:** 1Charles Perkins Centre, Faculty of Medicine and Health, School of Pharmacy, The University of Sydney, D17, The Hub, Camperdown, Sydney, NSW 2006, Australia; smcd4282@uni.sydney.edu.au (S.M.); lbar4259@uni.sydney.edu.au (L.B.); 2Centre for Health Systems and Safety Research, Australian Institute of Health Innovation, Faculty of Medicine, Health and Human Sciences, Macquarie University, Sydney, NSW 2109, Australia; 3School of Public Health and Preventative Medicine, Monash University, 553 St Kilda Road, Melbourne, VIC 3004, Australia; joanne.mckenzie@monash.edu.au (J.E.M.); matthew.page@monash.edu (M.J.P.); 4Charles Perkins Centre, Faculty of Science, School of Life and Environmental Sciences, The University of Sydney, D17, The Hub, Camperdown, Sydney, NSW 2006, Australia; margaret.allman-farinelli@sydney.edu.au (M.A.-F.); david.raubenheimer@sydney.edu.au (D.R.); 5School of Medicine and Colorado School of Public Health, Centre for Bioethics and Humanities, University of Colorado Anschutz Medical Campus, 13080 E. 19th Ave, Mail Stop B137, Aurora, CO 80045, USA; lisa.bero@cuanschutz.edu

**Keywords:** vitamin D, calcium, osteoporosis prevention, guideline development methods, evidence-based guidelines, public health

## Abstract

Background: There are numerous guidelines developed for bone health. Yet, it is unclear whether the differences in guideline development methods explain the variability in the recommendations for vitamin D and calcium intake. The objective of this systematic review was to collate and compare recommendations for vitamin D and calcium across bone health guidelines, assess the methods used to form the recommendations, and explore which methodological factors were associated with these guideline recommendations. Methods: We searched MEDLINE, EMBASE, CINAHL, and other databases indexing guidelines to identify records in English between 2009 and 2019. Guidelines or policy statements on bone health or osteoporosis prevention for generally healthy adults aged ≥40 years were eligible for inclusion. Two reviewers independently extracted recommendations on daily vitamin D and calcium intake, supplement use, serum 25 hydroxyvitamin D [25(OH)D] level, and sunlight exposure; assessed guideline development methods against 25 recommended criteria in the World Health Organization (WHO) handbook for guideline development; and, identified types identified types of evidence underpinning the recommendations. Results: we included 47 eligible guidelines from 733 records: 74% of the guidelines provided vitamin D (200~600–4000 IU/day) and 70% provided calcium (600–1200 mg/day) recommendations, 96% and 88% recommended vitamin D and calcium supplements, respectively, and 70% recommended a specific 25(OH)D concentration. On average, each guideline met 10 (95% CI: 9–12) of the total of 25 methodological criteria for guideline development recommended by the WHO Handbook. There was uncertainty in the association between the methodological criteria and the proportion of guidelines that provided recommendations on daily vitamin D or calcium. Various types of evidence, including previous bone guidelines, nutrient reference reports, systematic reviews, observational studies, and perspectives/editorials were used to underpin the recommendations. Conclusions: There is considerable variability in vitamin D and calcium recommendations and in guideline development methods in bone health guidelines. Effort is required to strengthen the methodological rigor of guideline development and utilize the best available evidence to underpin nutrition recommendations in evidence-based guidelines on bone health.

## 1. Introduction

Due to global aging, the prevalence and incidence of osteoporosis and fractures continue to rise in both developed and developing countries [1]. The social and economic burdens associated with osteoporosis, particularly fractures at the hip, are substantial, including disability, fracture recurrence, and premature mortality [2,3,4]. However, an effective and feasible non-pharmacological prevention strategy is yet to be widely endorsed. 

Vitamin D and calcium are two essential nutrients for normal bone growth and bone maintenance. Calcium plays a crucial role in skeletal mineralization, supporting bone strength and muscle contraction. Vitamin D facilitates calcium absorption via its active hormonal form, 1,25-dihydroxycholecalciferol [1,25(OH)2D3], working together with the parathyroid hormone to maintain calcium homeostasis [1,5]. There is no doubt that maintaining sufficient vitamin D and calcium is vital at every age. However, factors such as sunlight exposure, dietary habits, genetic and cultural backgrounds, and the aging process can contribute to different physiological needs for vitamin D and calcium [6,7,8,9]. To date, what is considered the appropriate level of vitamin D and calcium supplementation, as well as serum 25-hydroxycholecalciferol [25(OH)D] concentration, remains controversial [10].

In public health and clinical guidelines on bone health, vitamin D and calcium recommendations constitute an important, non-pharmacological strategy for bone maintenance and the prevention of osteoporosis and fractures. However, specific recommendations vary in the guidelines from different countries and even differ among organizations within a country. This is partially due to the conflicting evidence seen in the effectiveness of vitamin D and calcium supplements on bone mineral density [11,12] and fractures [13,14,15,16]. Furthermore, using these supplements at high doses may pose adverse events, including falls [17,18], cardiovascular diseases, and kidney stones [16,19]. 

Like other health conditions, bone health guidelines are ubiquitous in the healthcare system. Many national and international bodies have established similar standards to develop evidence-based guidelines [20,21,22,23]. For example, the World Health Organization (WHO) has provided standards for guideline development and recommends that guidelines should be developed by multidisciplinary committees or panels using a rigorous approach, including the systematic review of the available evidence and rating the strength of recommendations. The guideline development process should be transparent, including managing conflict of interest and tailoring to multiple stakeholders, such as policymakers, healthcare providers, patients, and the general public. Finally, guidelines need regular reviews and updates, in light of emerging evidence [20]. For evidence-based bone health guidelines, it is currently unclear to what extent the variability in vitamin D and calcium recommendations is related to the methods used to develop these recommendations. 

The objective of this study was to collate and compare recommendations for vitamin D and calcium across bone health guidelines globally, use the guideline development methods recommended by the WHO to appraise the quality of the methods used to develop guideline recommendations, and identify methodological factors that might affect the recommendations of vitamin D and calcium intakes, dietary and supplemental intakes, serum level of 25(OH)D, and sunlight exposure. 

## 2. Methods

We registered this systematic review in PROSPERO (registration number: CRD42019126452) in March 2019 and published a peer-reviewed protocol [24].

### 2.1. Data Sources and Searches

Working with an experienced academic librarian, we searched for bone health guidelines or policy statements in the following electronic databases: MEDLINE (via OVID), EMBASE (via OVID), CINAHL (via EBSCO), Practice-Based Evidence in Nutrition, National Guideline Clearinghouse (by Agency for Healthcare Research and Quality, AHRQ), NICE, and Guidelines International Network (GIN) in March 2019. The search period was restricted from 1 January 2009 to 28 February 2019. For other databases, the searches were based on a single keyword or a combination of vitamin D (or calcium), bone, osteoporosis, guideline/policy, and recommendation as search terms. The search strategy used to retrieve the guidelines in MEDLINE, EMBASE, and CINAHL as well as the other databases indexing guidelines mentioned above are described in the Appendix A. Additionally, we searched the website of the International Osteoporosis Foundation to capture missing guidelines or policy statements. 

### 2.2. Study (Guideline/Policy Statement) Selection

Details of the inclusion and exclusion criteria were published in our protocol [24]. To qualify as a guideline recommendation, we adopted the definition described in the 2014 WHO handbook for guideline development, that is, “any document containing recommendations for clinical practice or public health policy. A recommendation tells the intended end-user of the guideline what he or she can or should do in specific situations to achieve the best health outcomes possible, individually or collectively” [20]. We only included the most up-to-date bone health guidelines on the prevention of osteoporosis and fractures developed by a nationally or internationally recognized government authority, a medical/academic society, or an organization within a country or a special region (such as Hong Kong or Taiwan). Our target population was generally healthy adults aged 40 years and older who were at risk of developing osteoporosis. The reason for selecting this age group was that some women might experience early menopause as young as 40 years [25,26]. Due to the available resources, we only included guidelines written in English. We excluded bone health guidelines related to the management of osteoporosis (such as postmenopausal women under a physician’s care) and secondary osteoporosis (e.g., osteoporosis due to rheumatoid arthritis or glucocorticoid-induced osteoporosis). Also, we excluded guidelines targeted to a particular group of population or those with health conditions such as HIV, cancers, and guidelines on clinical treatments of any bone disorders [24]. As we focused on bone health guidelines, government reports on nutrient reference values (nutrient requirements at the population level) are out of scope in this review. Unlike bone health guidelines, government nutrient reference reports are developed based on the nutritional adequacy essential for normal physiological functioning (e.g., body stores of nutrients and enzymatic reactions), prevention of symptoms and conditions related to nutrient deficiency, chronic disease prevention, and risks of adverse effects from excessive intakes for the whole population. The determination of nutrient reference values uses methods including population surveys, dose-dependent or factorial approaches, human and animal studies, systematic reviews, and the consideration of the characteristics of dietary intake to determine dietary reference intakes [27,28,29]. This is a comprehensive process that is different from the development of a specific bone health guideline. 

One reviewer (ZD) screened the titles and abstracts of all the retrieved records. The full text of potentially eligible guidelines or policy statements was further assessed and finalized among three reviewers (ZD, SM, and LB) based on reading the full text thoroughly and the eligibility criteria noted above. 

### 2.3. Data Extraction and Quality Assessment

Data on guideline characteristics, vitamin D and calcium recommendations, and evidence cited to support the recommendations were extracted (Table 1). For both nutrients, the extraction of daily vitamin D (international unit (IU)/day or µg/day) and calcium (mg/day), the recommendations were extracted as they were reported in the guidelines. 

To ensure consistency in the extraction of the recommendations, which comprised of both quantitative (i.e., with numerical values) and qualitative (i.e., descriptive text) information, we adopted the criteria proposed by Woolf and colleagues [30]. We first extracted the evidence underpinning the recommendations, followed by retrieval of the full-text articles, which were then categorized by evidence type. Additionally, more than one evidence type could be selected per recommendation. Two reviewers (ZD and SM or LB) independently extracted the data and any discrepancies in the data extraction or categorizations were resolved via discussion or through consultation with the senior author (LAB). 

### 2.4. Assessment of Guideline Development Methods

Using a content analysis approach, two reviewers (ZD and SM or LB) independently appraised each guideline according to a set of 25 criteria, adopted from the 2014 WHO handbook for guideline development [20]. The WHO methodological criteria include those in the Appraisal of Guidelines for Research and Evaluation (AGREE) II [31] that are commonly used for assessing guideline quality and have additional considerations such as using systematic review methods to search, retrieve, and synthesize evidence; the transparency of the types of evidence used; the rating of the importance of the health outcome; and health equity, acceptability, and feasibility in the process of formulating recommendations [20]. Details of the criteria and our rationale for using the WHO guideline methods were described in our protocol [24]. 

For each guideline, we rated whether each of the recommended WHO methodological criteria were applied (Appendix A) using the response options Yes, No, or Unclear. If a method was used (response option “Yes”), verbatim text from the guideline (or Appendix A) was extracted. “No” was selected when a guideline explicitly stated that it did not adopt the specific process (e.g., “there was no stakeholder involvement in this guideline”). If the process was not explicitly stated as not being used or was not described, we used the response option “Unclear”. Any discrepancies for appraisal of the guideline development processes were resolved via discussion among the reviewers or consultation with the senior author (LAB).

All data extraction and method assessments of the guideline development processes were captured and stored using the Research Electronic Data Capture, an electronic data capture tool hosted at the University of Sydney [32]. The data were then exported to Excel for data cleaning and analysis in other statistical programs. 

### 2.5. Data Synthesis and Analysis

Descriptive summary statistics, using frequencies and percentages, were used to summarize guideline characteristics and the recommendations made for the daily recommended intake of vitamin D (IU/day; 1 μg = 40 IU), calcium (mg/day), and the recommended level of serum 25(OH)D (1 ng/mL = 2.5 nmol/L). We also recoded some of the recommendations into the dietary recommendations made for either vitamin D/calcium-rich food (Yes/No), supplement recommendations made for either vitamin D/calcium supplements (Yes/No), and sunlight exposure recommendation (Yes/No). For the types of evidence supporting the recommendations, we combined three groups of guidelines (source country of guideline, previous guidelines from other countries, and international guidelines (e.g., guidelines published from the World Health Organization)) into one category. 

For assessing guideline development methods, we calculated the mean (95% confidence interval, CI) of the WHO methodological criteria met for each guideline and the proportion (95% CI) of guidelines that fulfilled each of the criteria. We examined the association between each of the WHO methodological criteria and the forming recommendations of daily intake for vitamin D and calcium to investigate whether the variability in the quality of the guideline methods could potentially explain the variations in the recommendations. Because of the small sample of the guidelines, we combined different types of recommendations into a binary variable, i.e., vitamin D or calcium recommendation made (or not made). We calculated the risk difference (RD) between a guideline process that had met or not met a WHO guideline development criterion and the recommendations made on vitamin D/calcium (e.g., we compared guidelines that met “discipline representation” with those that did not and what the difference was for guidelines where a recommendation was made for daily intake of vitamin D/calcium). We used a confidence level of 99% to calculate the confidence intervals for these differences (rather than 95%) to facilitate greater caution in our interpretation of the results, due to the number of associations examined. We calculated Fisher’s exact P-value because there were a small number of guidelines. 

All analyses were performed using SAS (version 9.4) or Stata version 16 [33]. The forest plot was produced using the metaprop package [34]. A two-sided *p*-value (<0.05) is considered statistically significant. 

### 2.6. Role of the Funding Source

This study was supported by the Australian National Health and Medical Research Council (NHMRC) project grant (APP1139997), which aims to strengthen the evidence foundation for public health guidelines. The NHMRC is one of the two major government funding agencies that allocates competitive research funding at Australian universities. The funding source played no role in the conceptualization, design, analytical methods, data interpretation, reporting of the manuscript, and publication decisions.

## 3. Results

### 3.1. Results of Search

After the removal of 160 duplicates from the 733 records identified from different databases, 573 records remained. A further 456 records were removed after screening the title or abstract and 117 records underwent full-text screening, from which 47 eligible guidelines were included (Figure 1). Appendix A lists the guidelines included in this review and Appendix A lists the excluded documents in the review. 

### 3.2. Characteristics of Guidelines

Among the 47 guidelines, 65% were published between 2009 and 2014 (Table 2). Most guidelines (35, 74%) targeted adults aged 40 or 50 years and over; 26% (*n* = 12) were for women only. Based on the World Bank Gross National Income per capita [35], 81% (*n* = 35) were developed in countries at middle-upper or high-income levels; a majority of the guidelines originated in Europe (16, 34%) and North America (11, 23%). A medical or academic society produced most of the guidelines (42, 89%) and 62% (*n* = 29) did not disclose a funding source. 

### 3.3. Recommendations on Vitamin D and Calcium

The daily intake recommendations on vitamin D or calcium vary considerably in the guidelines. For vitamin D, most guidelines (37, 79%) provided a daily intake recommendation ranging from 200~600 IU to 4000 IU per day (Table 3). Among these guidelines, twelve specified the recommended intakes from nutrient reference values: eight guidelines adopted the Recommended Dietary Allowance (RDA) by the US Institute of Medicine [27], two adopted the Reference Nutrient Intakes (RNIs) by the UK Scientific Advisory Committee on Nutrition (SACN) [36], one adopted the Recommended Energy and Nutrition Intake (RENI) in the Philippines [37], and one cited the DRIs for vitamin D [27]. Additionally, these vitamin D recommendations varied by source: 40% of the guidelines (*n* = 19) recommended getting vitamin D from dietary sources, 96% of the guidelines (*n* = 45) recommended taking vitamin D supplements, and 26% of the guidelines (*n* = 12) recommended getting vitamin D from outdoor sunlight to maintain bone health. Notably, one guideline may recommend getting vitamin D from different sources; therefore, the percentages of the different types of sources do not add up to 100. Furthermore, 33 guidelines (70%) recommended serum concentration for 25(OH)D, ranging from 25 to 75~250 nmol/L (i.e., 10 to 30 ~100 ng/mL) and used words such as “optimal”, “sufficient”, “adequate”, “ideal”, “desirable”, “sustained”, “required”, and “minimal” for the recommended level. 

For calcium, 35 guidelines (74%) provided daily recommended intake, ranging from 600 to 1300 mg (Table 3). Among these guidelines, fourteen specified the recommended intake from nutrient reference values: eight were based on the RDA by the IOM [27], one was based on Adequate Intake (AI) by the IOM [38], three cited the RDA equivalence from their country’s reference values [37,39,40], and two were based on DRIs [27]. Most guidelines recommended getting calcium from specific food sources (e.g., dairy) (34, 72%) and supplements (38, 81%). 

Regarding the recommendations on a whole foods diet, only 17% (*n* = 8) of the guidelines recommended having a nutritionally balanced diet to maintain bone health or prevent osteoporosis. 

### 3.4. Assessment of Guideline Methods

On average, 10 (95% CI of the mean: 9–12; interquartile range: 6–15) of the 25 WHO methodological criteria were met per guideline. Four criteria that were met by more than 70% of the guidelines were: “Priority of the problem stated: Is the problem a burden of disease?” (83%), “Disclosure of conflicts of interest” obtained (74%), “Are recommendations explicitly linked to evidence?” (74%), and “Discipline representation” of the guideline development group (70%) (Figure 2). The least frequently met criteria included “Diversity representation” (4%), “Used systematic review methods to synthesize evidence” (11%), “Conflicts of interest managed” (15%), “Outcome importance specified—uncertainty about or variability in how much people value the main outcome?” (19%), “Health equity” considered (21%), “Acceptability” considered (21%), and external review of guidelines conducted (23%). Detailed descriptions of the WHO guideline development methods criteria are available in Appendix A. In the top 10 guidelines that met at least 16 of the 25 WHO recommended methodological criteria, the daily vitamin D recommended intakes ranged from 160 IU–1000 IU and the daily calcium recommendations were between 700 and 1200 mg/day (Appendix A). 

There was no clear evidence of an association between the WHO methodological criteria being met and making recommendations for daily intake of vitamin D or calcium (Table 4). The CIs for the differences of the proportion of guidelines met a criterion versus those that did not meet the criterion for the relationship of making the recommendations were generally wide, often including potentially important differences at either end of the confidence limits. Even though we found statistically significant results between “Acceptability of recommendations to stakeholders” and recommendations made for the daily intake of vitamin D (RD: −0.38, 99% CI (−0.8, 0.04), and *p* = 0.02) and between “guidelines that had undergone an external review” and recommendations made for daily intake of calcium (RD: 0.34, 99% CI: 0.14, 0.55, and *p* = 0.04), we cannot rule out whether these results were found by chance, due to the small sample size and wide confidence intervals. 

### 3.5. Evidence Cited to Support Recommendations

Previous bone health guidelines were cited the most to support the recommendations, except for those on sunlight exposure (commentary) and supplemental calcium (systematic reviews of RCTs) (Table 5). For example, 57% (20/35) of the guidelines cited previous guidelines for making recommendations on serum 25(OH)D concentration and 47% (21/45) on vitamin D supplementation. Among those cited previous guidelines on daily intake of vitamin D and calcium, 12 cited government reference values to support daily vitamin D intake and 14 did so to support daily calcium intake, along with citing previous bone health guidelines. Notably, systematic reviews of RCTs were cited in only 8–42% of the guidelines supporting individual recommendations for vitamin D or calcium. Narrative reviews, editorials, or commentaries (33%) constituted the primary supporting evidence for sunlight exposure.

## 4. Discussion 

In this systematic review of 47 evidence-based bone health guidelines around the world, we found that the range of the recommended daily intake and supplementation of vitamin D, calcium, and serum level of 25(OH)D varied substantially; most of the guidelines recommended taking supplements for bone health and the prevention of osteoporosis. Overall, the methodological quality of these guidelines is low, with, on average, only 10 out of 25 of the WHO methodological criteria for guideline development being met. Particularly concerning criteria included, lack of diversity of representation in the guideline development group, little description of the management of conflict of interest (COI), limited use of systematic review methods to synthesize evidence, lack of consideration in health equity and acceptability to meet target users’ needs, and there was a paucity of external review of the guidelines. With a small sample of bone health guidelines, although we could not find clear evidence of the associations between the WHO methodological criteria and variabilities of the recommendations on daily intake of vitamin D or calcium, we found that the primary source of evidence underpinning most recommendations was previously published guidelines. 

The substantial variability in the recommended levels of vitamin D (200~600–4000 IU/day) and calcium (600–1300 mg/day), particularly vitamin D, and a lack of systematic reviews as supporting evidence, raises concerns about vitamin D and calcium supplementation recommended by these evidence-based bone health guidelines. As a comparison, the WHO nutrient requirement intake (NRI) for vitamin D is 200 IU/day for adults aged 19–50 years, 400 IU/day for those aged 51–65 years, 600 IU/day for those aged 65+ years, 1000 mg/day for 19–65 years/menopause, and 1300 mg/day for 65 years+/post menopause [42]. The US DRIs provide an RDA (covering 97.5% of the population) for calcium as 1000 mg/day for men 31–70 years and women 31–50 years and 1200 mg/day for men over 70 years and women over 50 years; for vitamin D the provided RDA is 600 IU/day for adults 31–70 years and 800 IU/day for those over 70 years [27]. In the more recent recommendations on vitamin D and calcium published in 2017 by the European Food Safety Authority, the Adequate Intake for vitamin D is 600 IU (15 ug)/day for adults aged 18 years and above to achieve 50 nmol/L 25(OH)D in the majority of the population [43] and 950 mg/day calcium as a Population Reference Intake for adults aged 25 years and above [44]. Therefore, vitamin D recommendations in the sample of bone health guidelines assessed vary considerably. 

Moreover, current findings on the effectiveness of vitamin D and calcium supplementation on bone mineral density and on fracture prevention are inconsistent. Several systematic reviews of RCTs have suggested no beneficial effect of vitamin D supplementation on the prevention of total fractures or hip fractures [11,12,13,45,46] or on BMD [11,12,14]. Two large trials published in 2019 with over 12 months of follow-up also found no effects of vitamin D supplementation on bone health [18,47]. Likewise, previous reviews have raised issues regarding the effectiveness of calcium supplementation on BMD and fracture prevention [10,16,48]. The effects of vitamin D in combination with calcium supplements on reducing non-vertebral fractures were also inconsistent in systematic reviews [45,46]. Additionally, adverse events such as hypercalcemia and hypercalciuria [47], fall risks [18], cardiovascular events [16,19], gastrointestinal symptoms, and renal disease [45] have been reported in systematic reviews of clinical trials. Although quite a few of the guidelines in this review were developed more than five years ago and the evidence underpinning the recommendations on vitamin D and calcium was not the most recent, results from several meta-analyses of clinical trials published before and after 2010 have suggested that the effect of vitamin D or calcium supplements on bone mineral density or fracture prevention is uncertain [12,14,19,48] and that the point estimates for supplements of vitamin D with or without calcium for fracture risk did not change materially since before 2010 when these guidelines were released [12]. As noted in Table 5, relatively few of the guidelines cited RCTs, or systematic reviews of RCTs, and we cannot be certain that this was due to the availability of RCTs or the methods used by the guideline developers. Along with the low methodological quality of these guidelines, health professionals should be cautious regarding the prescribed daily dose of vitamin D or calcium supplements to individuals at risk of developing osteoporosis or fractures. 

With a sample of 47 guidelines and the width of confidence intervals from our results, we were not able to establish whether any of the guideline processes were associated with the recommendations made on daily intake of vitamin D or calcium, nor the variation of the recommended levels. However, the top 10 guidelines that met at least 16 of the 25 WHO methodological criteria (Appendix A) for guideline development show comparable ranges with the reference values listed in the government authorities mentioned above [29,42,43,44]. As various physiological factors [6,7,8,9], as well as cultural and religious practices [49], can affect the nutritional needs of vitamin D and calcium in different populations, efforts including increased stakeholder (target users) inputs and an external peer review process could prompt guideline committees to consider these factors. 

Although most guidelines disclosed authors’ COI, only 15% described a procedure to manage it. Disclosure alone does not prevent bias associated with COI; hence, any identified COI must be eliminated or managed [50,51] to reduce the risk of influencing recommendations [52]. Likewise, transparency of the disclosure and management of COI in bone health guidelines may reduce these potential biases in the recommendations and increase credibility.

We also found that many bone health guidelines used previous bone guidelines to support their recommendations. A similar finding was reported in a review of national dietary guidelines in support of dietary recommendations [53]. The high reliance on previous guidelines could be due to the limited capacity of some countries to invest in evidence synthesis for guideline development. While referencing nutrient reference reports that were rigorously developed (such as those by the IOM [27], SCAN [36,39], and WHO [42]) should be encouraged, relying on previous bone health guidelines as the primary evidence base with a lack of systematic reviews to support the recommendations may create the risk of perpetuating inadequate guideline development methods and evidence [53]. Additionally, evidence-to-decision frameworks exist, such as the WHO-INTEGRATE framework [54]; these were not mentioned as methods used for recommendation formulation. 

This review has several strengths. We adopted a comprehensive search strategy to identify global bone health guidelines and policy statements related to vitamin D and calcium recommendations among generally healthy adults aged 40 years and above. Our research questions, search strategy, and planned methods were developed a priori and published in a peer-reviewed journal [24]. Two reviewers independently appraised guideline development methods using the 2014 WHO Handbook for Guideline Development a “gold standard” for developing public health and clinical guidelines, in the context of global populations. Also, recommendations were independently extracted under standard guidance [30]. 

This review also has limitations. We primarily relied on what was documented in the published guidelines. This may limit our ability, for criteria marked as “Unclear”, to fully determine whether a process was not implemented or simply not mentioned. Secondly, certain aspects of the development processes were deemed missing if guideline development standards other than the WHO methodological criteria were used. Thirdly, although we assessed whether systematic review methods were adopted to identify and evaluate evidence for each guideline, we did not evaluate whether an original systematic review was conducted or whether previous guidelines were the primary source of evidence. Also, we did not assess the methodological rigor of previous guidelines that were used as a primary source of evidence. Including English only publications has limited our sample size and regional coverage, resulting in a higher portion of guidelines from English-speaking countries. 

In conclusion, we found that recommendations on vitamin D and calcium vary substantially in a set of 47 evidence-based bone health guidelines across different countries/regions in the world; variability exists in guideline development methods and in the types of evidence underpinning the recommendations. From the public health’s perspective, these substantial variations in the bone health guidelines could affect clinicians’ prescribing decisions regarding vitamin D/calcium supplemental intake levels for their patients to prevent osteoporosis. Hence, our review points to the importance of adhering to the standard evidence-based guideline development methods to formulate rigorous recommendations.

In summary, our findings suggest that future guideline groups should document efforts to utilize the best available evidence to substantiate nutritional recommendations, implement procedures to eliminate or manage potential conflicts of interest, and address health equity and acceptability in formulating guideline recommendations. This review provides a benchmark of methods and processes used to develop public health guideline recommendations, against which future studies can be compared. 

## Figures and Tables

**Figure 1 nutrients-13-02423-f001:**
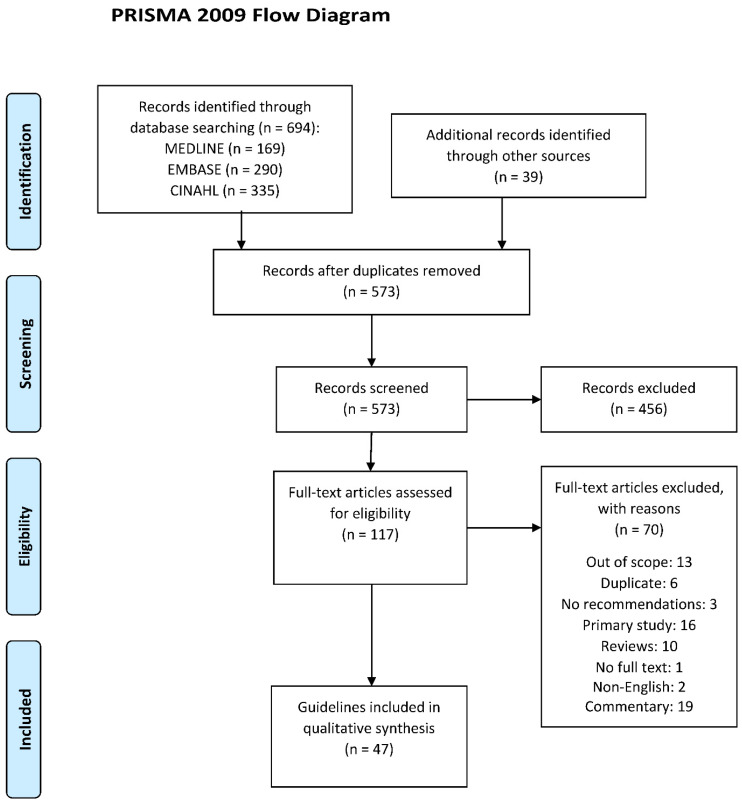
PRISMA to summarize guidelines/policy statements search and selection (https://guides.lib.unc.edu/prisma, accessed on 20 November 2019).

**Figure 2 nutrients-13-02423-f002:**
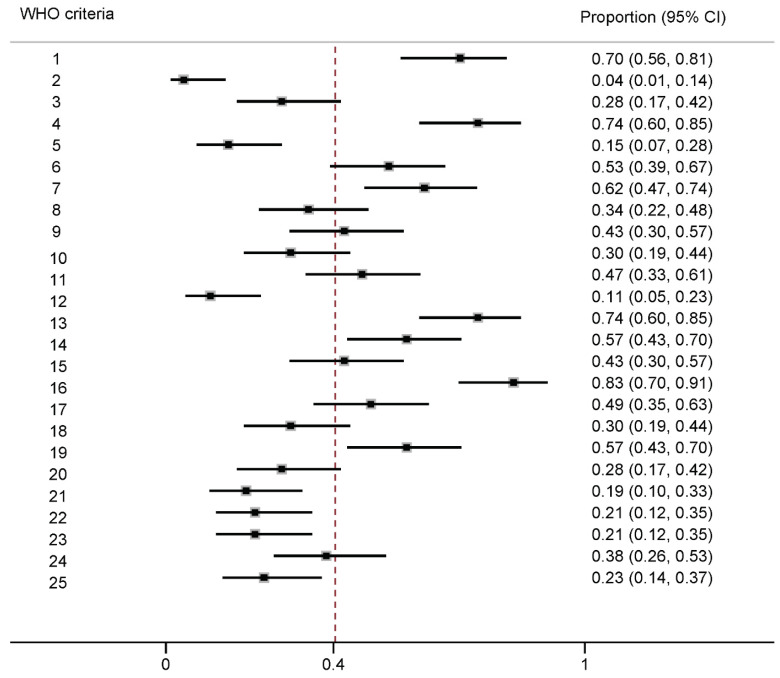
Estimated proportion (95% confidence interval) of guidelines that fulfilled each of the 25 WHO methodological criteria ^1^ for guideline development. ^1^ WHO methodological criteria [20]: 1. Discipline representation; 2. Diversity representation; 3. Stakeholder input; 4. Disclosure of conflicts of interest obtained; 5. Conflicts of interest managed; 6. Disclosure of funders of the guideline obtained and disclose funder’s role in influencing the guideline development process and recommendations; 7. Formulation of key questions for the evidence review in PICO, PICOT, or PEO format; 8. Choosing (finalizing) priority outcomes for systematic review; 9. Used systematic methods to search for evidence; 10. Used systematic methods to retrieve evidence to select eligible studies; 11. Used systematic methods to assess quality of evidence; 12. Used systematic methods to synthesize evidence; 13. Are recommendations explicitly linked to evidence?; 14. Was a consensus process clearly described for developing recommendations?; 15. Was a method employed to determine strength and/or certainty of the recommendation?; 16. Priority of the problem: Is the problem a burden of disease? 17. Quality of the evidence: Is higher quality of the body of evidence included to support the recommendation?; 18. Certainty of evidence: Does the recommendation include a consistent body of evidence?; 19. Benefits and harms: Are evaluations performed on the net benefit or net harm associate with an intervention or exposure?; 20. Balance: Does the balance between desirable and undesirable effects support the recommendation?; 21. Outcome importance: Is there important uncertainty about or variability in how much people value the main outcome?; 22. Equity: Does the evidence used reduce inequalities, improve equity, or contribute to the realization of one of several human rights defined under the international legal framework?; 23. Acceptability: Is the option acceptable to key stakeholders?; 24. Feasibility: Is the option feasible to implement?; 25. Was the guideline/recommendation reviewed by an external review group?

**Table 1 nutrients-13-02423-t001:** Data items extracted from guidelines.

Characteristics of Guidelines	Recommendations on Vitamin D and Calcium	Evidence Cited to Support Recommendations
Title; Guideline developing authority or organization; Publication year; Age group; Sex of target population; Funding source	Daily intake of vitamin D (IU/day); Dietary intake of vitamin D rich foods; Dosage of vitamin D supplements (IU/day); Daily intake of calcium (mg/day); Dietary intake of calcium rich foods; Dosage of calcium supplements (mg/day); Sunlight exposure; Recommended level of serum 25(OH)D (nmol/L)	Verbatim text that referenced the supporting studies/guidelines; Full-text articles of supporting studies grouped to the following types (previous guidelines from other countries): International guidelines (e.g., guidelines published from the World Health Organization; Systematic review of RCTs; Systematic review of non-randomized studies; Clinical trial; Cohort study; Cross-sectional study; Case-control study; Other (such as narrative review, editorial, and commentary)

**Table 2 nutrients-13-02423-t002:** Characteristics of included guidelines (*n* = 47).

Characteristics	*N*	Percentage (%)
Year of publication		
2009–2014	31	65
2015–2019	16	35
Age distribution		
≥40 years	13	27
≥50 years	22	47
≥60 years	6	13
Adults in general (≥18 years)	6	13
Sex distribution		
Women	12	26
Men	1	2
Both sexes	34	72
WHO regions ^1^		
Africa	1	2
Americas	13	28
South-East Asia	2	4
Europe	16	34
Eastern Mediterranean	1	2
Western Pacific	12	26
Other (international guidelines)	2	4
Guideline organization category		
Medical or academic society	42	89
Government body	5	11
Funding source		
None disclosed	29	62
Government agency	8	17
Medical society	5	11
Pharmaceutical/Food Industry	4	9
Mixed sources	1	2

^1^ World Health Organization. Definition of regional groupings 2020 [41].

**Table 3 nutrients-13-02423-t003:** Percentage of guidelines presenting recommendations on vitamin D and calcium (*n* = 47).

Recommendations	Counts	Percentage of Guidelines
Vitamin D		
Recommended daily intake (IU/d)	37	79%
200–600	1	3%
400–800	3	8%
400–800; 800–1000 ^1^	1	3%
400	3	8%
800	6	16%
600; 800 ^1^	1	3%
600–800	4	11%
800–1000	8	22%
800–2000	4	11%
>800	2	5%
1000	1	3%
1000–2000	1	3%
1500–2000	1	3%
4000	1	3%
Recommendation on dietary vitamin D	19	40%
Recommendation on supplemental vitamin D	45	96%
Recommendation on sunlight exposure	12	26%
Recommended level of serum 25(OH)D concentration (nmol/L)	33	70%
>25	1	2%
50	3	6%
75	9	19%
80	1	2%
≥70	1	2%
≥75	3	6%
>50	1	2%
>60	1	2%
50–125; 75–200 ^1^	1	2%
50–75	4	9%
50–80	1	2%
68–75	1	2%
75–125	4	9%
75–150	1	2%
75–250	1	2%
Calcium		
Recommended daily intake (mg/d)	35	74%
600	1	3%
750	1	3%
700; 800 ^1^	1	3%
700–1200	2	6%
800–1000	2	6%
800–1200	1	3%
1000	4	11%
1200	14	40%
1000; 1200 ^1^	3	9%
1000; 1300 ^1^	1	3%
1000–1200	5	14%
Recommendation on dietary calcium	34	72%
Recommendation on supplemental calcium	38	81%
For both nutrients		
Recommendation on a whole foods diet to maintain bone health	8	17%

^1^ represents separate recommendations exist in the same guideline for different age groups.

**Table 4 nutrients-13-02423-t004:** Risk difference (RD) and its 99% confidence interval (CI) for the relationship between a guideline process that had met or not met a WHO guideline development criterion ^1^ and the recommendations made on vitamin D or calcium.

WHO Guideline Development Criteria	Recommendation Made for Daily Intake of Vitamin D		Recommendation Made for Daily Intake of Calcium	
	RD (99% CI)	*p*-Value	RD (99% CI)	*p*-Value
1. Discipline representation	−0.16 (−0.48, 0.16)	0.30	0.04 (−0.32, 0.41)	0.73
2. Diversity representation	−0.26 (−1.18, 0.67)	0.45	−0.26 (−1.18, 0.67)	0.45
3. Stakeholder input	−0.15 (−0.52, 0.23)	0.47	−0.04 (−0.41, 0.32)	0.73
4. COI disclosure	−0.01 (−0.38, 0.37)	1.00	−0.12 (−0.46, 0.22)	0.70
5. COI managed	−0.04 (−0.51, 0.44)	1.00	0.13 (−0.25, 0.52)	0.66
6. Funder disclosure	0.03 (−0.3, 0.36)	1.00	0.03 (−0.3, 0.36)	1.00
7. PICO format of research question	−0.03 (−0.37, 0.31)	1.00	0.06 (−0.29, 0.41)	0.73
8. Priority outcomes	0.03 (−0.31, 0.37)	1.00	0.03 (−0.31, 0.37)	1.00
9. Systematic search	0.18 (−0.13, 0.49)	0.19	0.18 (−0.13, 0.49)	0.19
10. Systematic review methods to retrieve evidence	0.06 (−0.29, 0.4)	1.00	0.06 (−0.29, 0.4)	1.00
11. Evidence quality assessment	−0.03 (−0.36, 0.3)	1.00	−0.12 (−0.45, 0.21)	0.51
12. Systematic review methods to synthesize evidence	0.06 (−0.43, 0.55)	1.00	0.06 (−0.43, 0.55)	1.00
13. Recommendations linked to evidence	−0.1 (−0.46, 0.25)	0.70	−0.1 (−0.46, 0.25)	0.70
14. Consensus process	−0.01 (−0.34, 0.32)	1.00	0.08 (−0.26, 0.41)	0.74
15. Method employed to determine strength and certainty of recommendations	0.01 (−0.32, 0.34)	1.00	0.01 (−0.32, 0.34)	1.00
16. Priority of problem of the disease	0.29 (−0.19, 0.78)	0.18	−0.01 (−0.44, 0.43)	1.00
17. Quality of evidence	0.07 (−0.25, 0.4)	0.74	−0.01 (−0.34, 0.32)	1.00
18. Certainty of evidence	0.06 (−0.29, 0.4)	1.00	0.06 (−0.29, 0.4)	1.00
19. Benefits and harms of recommendation	−0.25 (−0.55, 0.04)	0.09	0.01 (−0.32, 0.35)	1.00
20. Balance of desirable and undesirable effects of recommendations	−0.04 (−0.41, 0.32)	0.73	−0.04 (−0.41, 0.32)	0.73
21. Outcome importance of disease	0.07 (−0.31, 0.45)	1.00	0.2 (−0.11, 0.51)	0.41
22. Health equity	−0.02 (−0.42, 0.37)	1.00	0.1 (−0.26, 0.45)	0.70
23. Acceptability	−0.38 (−0.8, −0.04)	0.02	−0.14 (−0.56, 0.27)	0.44
24. Feasibility	−0.01 (−0.35, 0.32)	1.0	−0.01 (−0.35, 0.32)	1.00
25. External review of guideline/recommendations	0.1 (−0.25, 0.46)	0.70	0.34 (0.14, 0.55)	0.04

^1^ The 2014 World Health Organization (WHO) Handbook for Guideline Development criteria [20]: 1. Discipline representation; 2. Diversity representation; 3. Stakeholder input; 4. Disclosure of conflicts of interest obtained; 5. Conflicts of interest managed; 6. Disclosure of funders of the guideline obtained and disclose funder’s role in influencing the guideline development process and recommendations; 7. Formulation of key questions for the evidence review in PICO, PICOT, or PEO format; 8. Choosing (finalizing) priority outcomes for systematic review; 9. Used systematic methods to search for evidence; 10. Used systematic methods to retrieve evidence to select eligible studies; 11. Used systematic methods to assess quality of evidence quality; 12. Used systematic methods to synthesize evidence; 13. Are recommendations explicitly linked to evidence?; 14. Was a consensus process clearly described for developing recommendations; 15. Was a method employed to determine strength and/or certainty of the recommendation?; 16. Priority of the problem: Is the problem a burden of disease?; 17. Quality of the evidence: Is higher quality of the body of evidence included to support the recommendation?; 18. Certainty of evidence: Does the recommendation include consistent body of evidence?; 19. Benefits and harms: Are evaluations performed on the net benefit or net harm associate with an intervention or exposure?; 20. Balance: Does the balance between desirable and undesirable effects support the recommendation?; 21. Outcome importance: Is there important uncertainty about or variability in how much people value the main outcome?; 22. Equity: Does the evidence used reduce inequalities, improve equity, or contribute to the realization of one of several human rights defined under the international legal framework?; 23. Acceptability: Is the option acceptable to key stakeholders?; 24. Feasibility: Is the option feasible to implement?; 25. Was the guideline/recommendation reviewed by an external review group?

**Table 5 nutrients-13-02423-t005:** Number and percentage of guidelines (*n* = 47) citing different types of evidence in support of the recommendations.

Recommendations	Types of Evidence											
	Number of Guidelines Citing Evidence ^1^	Any Type of Previous Guideline ^2^	Source Country of Guideline ^2^	Previous Guideline from Other Countries^2^	WHO/FAO Guideline ^2^	Systematic Review of RCTs ^2^	Systematic Review of non-RCTs ^2^	Clinical Trial ^2^	Cohort Study ^2^	Cross-Sectional Study ^2^	Case-Control Study ^2^	Other ^2^ (Narrative Review, Editorial, Commentary)
**Vitamin D recommendations**												
Recommended daily intake	37 (79)	18 (49)	10 (27)	9 (24)	5 (14)	11 (30)	0 (0)	4 (11)	0 (0)	1 (3)	0 (0)	0 (0)
Dietary intake	19 (40)	7 (37)	3 (16)	4 (21)	0 (0)	4 (21)	0 (0)	1 (5)	0 (0)	1 (5)	0 (0)	1 (5)
Supplemental intake	45 (96)	21 (47)	12 (27)	10 (22)	4 (9)	19 (42)	2 (4)	14 (31)	3 (7)	2 (4)	2 (4)	6 (13)
Serum vitamin D level	33 (70)	20 (61)	10 (30)	12 (36)	6 (18)	11 (33)	4 (12)	3 (9)	5 (15)	7 (21)	0 (0)	5 (15)
Sunlight exposure	12 (26)	3 (25)	2 (17)	1 (8)	0 (0)	1 (8)	0 (0)	1 (8)	0 (0)	0 (0)	0 (0)	4 (33)
**Calcium recommendations**												
Recommended daily intake	35 (74)	17 (49)	10 (29)	6 (17)	2 (6)	10 (29)	0 (0)	8 (23)	2 (6)	2 (6)	0 (0)	4 (11)
Dietary intake	34 (72)	14 (41)	11 (32)	3 (9)	1 (3)	7 (21)	1 (3)	4 (12)	4 (12)	2 (6)	0 (0)	3 (9)
Supplemental intake	38 (81)	8 (21)	6 (16)	3 (8)	1 (3)	15 (41)	0 (0)	7 (19)	2 (5)	0 (0)	1 (3)	3 (8)

All numbers are presented as number (%) of guidelines; the proportion in percentage is not mutually exclusive. ^1^ The denominator is based on the 47 guidelines analyzed. ^2^ The denominator is based on the number of guidelines citing the evidence for a particular recommendation seen in the second column.

## Data Availability

Detailed search strategies and the included and excluded records are published in the Appendix A.

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
