# Peer review of "Assessment of the Methods Used to Develop Vitamin D and Calcium Recommendations—A Systematic Review of Bone Health Guidelines"

_nutrients, 2021, doi:10.3390/nu13072423_

Round 1
Reviewer 1 Report
Taking up this topic is very appropriate. I have no remarks to the methodology.
Minor comments:
In the paragraph „Recommendations on vitamin D and calcium”, the sentence "Additionally, these vitamin D recommendations varied by source: 40% (n = 19) from diet, 96% (n = 45) vitamin D supplements, and 26% (n = 12) from outdoor sunlight ”is not understandable, because the given percentage does not add up to 100%. What does it mean that 40% of the cited recommendations point to diet and 96% to vitamin D supplements? The reader may also understand that 40% of vitamin D supplied to the body must come from the diet. This should be made more explicit.
Is the WHO nutrient requirement intake for vitamin D for adults (200 IU/day) mentioned in the paragraph Discussion still is up to date (document from 2004 is cited here) ??? According to the European Food Safety Authority recommendation from 2017, Adequate Intake for vitamin D is 600 IU (15 µg). It should be emphasized that the lack of compliance as to the body's need for vitamin D among the such important institutions responsible for public health is incomprehensible and against this background it is hardly surprising that the recommendations on vitamin D in the guidelines varied substantially.
Overall, the article confirms that despite the fact that the role of vitamin D and calcium in maintaining bone health is most often emphasized in the literature on these ingredients, there is still no reliable knowledge about the body's need for vitamin D and calcium, as well as the proper concentration of vitamin D in the blood (range from 25 to 250 nmol / L). It is not favorable for physicians who have to make their own decisions about possible supplementation and the dose of this nutrients in patients. The same situation (lack of consensus) currently applies to the role of vitamin D in the prevention and treatment of COVID-19, but in this case it is only one year of research.
Author Response
July 8, 2021
Dear Reviewer,
Thank you for reviewing our manuscript, titled “Assessment of the methods used to develop vitamin D and calcium recommendations – a systematic review of bone health guidelines” to be considered for publication in Nutrients.
We appreciate the comments and have addressed them point by point and incorporated them in our revised manuscript. We hope that our revised manuscript has improved its clarity and quality.
Sincerely,
Zhaoli Dai and Lisa Bero
Reviewer 1:
Minor comments:
In the paragraph, Recommendations on vitamin D and calcium", the sentence "Additionally, these vitamin D recommendations varied by source: 40% (n = 19) from diet, 96% (n = 45) vitamin D supplements, and 26% (n = 12) from outdoor sunlight " is not understandable, because the given percentage does not add up to 100%. What does it mean that 40% of the cited recommendations point to diet and 96% to vitamin D supplements? The reader may also understand that 40% of vitamin D supplied to the body must come from the diet. This should be made more explicit.
Response: Thank you for pointing this out. We overlooked this sentence. To make it clear, we have revised the wording and explained why the percentage does not add up to 100%. It now reads on page 7: "Additionally, these vitamin D recommendations varied by source: 40% of the guidelines (n=19) recommended getting vitamin D from dietary sources, 96% of the guidelines (n=45) recommended taking vitamin D supplements, and 26% of the guidelines (n=12) recommended getting vitamin D from outdoor sunlight to maintain bone health. To be noted, one guideline may recommend getting vitamin D from different sources; therefore, the percentages of different types of sources do not add to 100."
Is the WHO nutrient requirement intake for vitamin D for adults (200 IU/day) mentioned in the paragraph Discussion still is up to date (document from 2004 is cited here) ??? According to the European Food Safety Authority recommendation from 2017, Adequate Intake for vitamin D is 600 IU (15 µg). It should be emphasized that the lack of compliance as to the body's need for vitamin D among the such important institutions responsible for public health is incomprehensible and against this background it is hardly surprising that the recommendations on vitamin D in the guidelines varied substantially.
Response: The 2004 WHO nutrient requirement intake for vitamin D is the most current document published by the WHO. We cited this document for providing an intake level of vitamin D and calcium oriented toward the global populations. Additionally, we cited the most recent US DRI for vitamin D and calcium as a comparison. Taking the reviewer's suggestion, we have added the recommendations from the European Food Safety Authority in this paragraph. It now reads, "In the more recent recommendations on vitamin D and calcium published in 2017 by the European Food Safety Authority, the Adequate Intake for vitamin D is 600 IU (15ug)/day for adults aged 18 years and above to achieve 50noml/L 25(OH)D in the majority of the population (43) and 950mg/day calcium as a Population Reference Intake for adults aged 25 years and above (44)." on page 9.
Although we agree with the reviewer, compliance with guideline recommendations is beyond the scope of this paper, and we have not stated our opinion on this point.
Overall, the article confirms that despite the fact that the role of vitamin D and calcium in maintaining bone health is most often emphasized in the literature on these ingredients, there is still no reliable knowledge about the body's need for vitamin D and calcium, as well as the proper concentration of vitamin D in the blood (range from 25 to 250 nmol / L). It is not favorable for physicians who have to make their own decisions about possible supplementation and the dose of this nutrients in patients. The same situation (lack of consensus) currently applies to the role of vitamin D in the prevention and treatment of COVID-19, but in this case it is only one year of research.
Response: Thank you for this summary. We have added comments about the variabilities in the vitamin D and calcium recommendations in these bone guidelines that could affect prescribing decisions on page 10. Now it reads, "Furthermore, these substantial variations in the bone health guidelines could affect clinicians' prescribing decisions on vitamin D/calcium supplemental intake levels to their patients for the prevention of osteoporosis."
Reviewer 2 Report
In this review Dai et al. aim to collate and compare recommendations for vitamin D and calcium in different bone health guidelines, assess the methods used, and explore methodological factors associated with these recommendations. The topic is interesting and data sources and searches are well structured.
Nevertheless, there are some minor points to address.
Some details of the protocol, such as inclusion and exclusion criteria, though well detailed elsewhere, should be included in this review, instead of description of government nutrient reference report.
Despite the great care in the search methodology, it seems that the conclusions are not up to the effort. Indeed, the considerable variability in vitamin D and calcium recommendations is a quite expected result, as stated in the abstract background.
In detail, the authors should better point out which recommendations are more reliable and which guidelines should be taken in more consideration among those evaluated. A table should be helpful for this scope.
Author Response
July 8, 2021
Dear Reviewer,
Thank you for reviewing our manuscript, titled “Assessment of the methods used to develop vitamin D and calcium recommendations – a systematic review of bone health guidelines” to be considered for publication in Nutrients.
We appreciate the comments and have addressed them point by point and incorporated them in our revised manuscript. We hope that our revised manuscript has improved its clarity and quality.
Sincerely,
Zhaoli Dai and Lisa Bero
Reviewer 2:
In this review Dai et al. aim to collate and compare recommendations for vitamin D and calcium in different bone health guidelines, assess the methods used, and explore methodological factors associated with these recommendations. The topic is interesting and data sources and searches are well structured.
Nevertheless, there are some minor points to address.
Some details of the protocol, such as inclusion and exclusion criteria, though well detailed elsewhere, should be included in this review, instead of description of government nutrient reference report.
Response: As suggested, we have added more details on the inclusion and exclusion criteria as stated in our published protocol to Methods, on page 3-4.
Despite the great care in the search methodology, it seems that the conclusions are not up to the effort. Indeed, the considerable variability in vitamin D and calcium recommendations is a quite expected result, as stated in the abstract background.
Response: We agreed with the reviewer on this point. However, as also noted in our Abstract, the objective of our study was to "explore which methodological factors were associated with these guideline recommendations." In other words, guidelines using different methods but not those described in the guideline development standards could come to different recommendations, even though they use the same evidence or not (e.g. grading the quality and certainty of the evidence). We have added this comment on page 10-11.
In detail, the authors should better point out which recommendations are more reliable and which guidelines should be taken in more consideration among those evaluated. A table should be helpful for this scope.
Response: As suggested, we included the top 10 guidelines that met at least 16 of the 25 methodological criteria recommended in the WHO Handbook and their recommendations on vitamin D and calcium in Supplemental Table 4 on page 15-18 in Supplemental materials. We added the brief results on page 7 in Results and comments on page 9 in Discussion to provide some evidence on the link of methodological rigor with recommendations on vitamin D and calcium intakes.
Reviewer 3 Report
The manuscript is a pains-taking review of the current guidelines and identifies their inadequacy and the need to develop coherent evidence-based set of guidelines.
Minor clarifications:
Abstract: The sentence “The mean of meeting 25 WHO methodological criteria per guideline was 10 (95% CI: 9-12; interquartile range: 6-15).” could be stated more clearly.
Figure1 and 2: I see the legends but not the figures
Author Response
July 8, 2021
Dear Reviewer,
Thank you for reviewing our manuscript, titled “Assessment of the methods used to develop vitamin D and calcium recommendations – a systematic review of bone health guidelines” to be considered for publication in Nutrients.
We appreciate the comments and have addressed them point by point and incorporated them in our revised manuscript. We hope that our revised manuscript has improved its clarity and quality.
Sincerely,
Zhaoli Dai and Lisa Bero
Reviewer 3:
The manuscript is a pains-taking review of the current guidelines and identifies their inadequacy and the need to develop coherent evidence-based set of guidelines.
Minor clarifications:
Abstract: The sentence "The mean of meeting 25 WHO methodological criteria per guideline was 10 (95% CI: 9-12; interquartile range: 6-15)." could be stated more clearly.
Response: We have changed the wording to make this clearer. It now reads: "On average, each guideline met 10 (95% CI: 9-12) of the total of 25 methodological criteria for guideline development recommended by the WHO Handbook." on page 2 in the Abstract.
Figure1 and 2: I see the legends but not the figures
Response: The figures were included in the figure folders. Nevertheless, we have inserted Figure 1 and Figure 2 in the main documents on page 24-26.